# Synthesis, Structure, and Spectroscopic Properties of Luminescent Coordination Polymers Based on the 2,5-Dimethoxyterephthalate Linker

**Aimée E. L. Cammiade** [iD]**, Laura Straub** [iD]**, David van Gerven, Mathias S. Wickleder** [iD] **and Uwe Ruschewitz** *[iD]

Department of Chemistry, University of Cologne, Greinstraße 6, D-50939 Cologne, Germany;
aimee.cammiade@uni-koeln.de (A.E.L.C.)
* Correspondence: uwe.ruschewitz@uni-koeln.de

**Abstract:** We report on the synthesis and the crystal structure of the solvent-free coordination polymer $Co^{II}(2,5\text{-DMT})$ (**1**) with 2,5-DMT ≡ 2,5-dimethoxyterephthalate which is isostructural to the already reported $Mn^{II}$ and $Zn^{II}$ congeners ($C2/c$, $Z = 4$). In contrast, for M = $Mg^{II}$, a MOF with DMF-filled pores is obtained, namely $Mg_2(2,5\text{-DMT})_2(DMF)_2$ (**2**) ($P\bar{1}$, $Z = 2$). Attempts to remove these solvent molecules to record a gas sorption isotherm did not lead to meaningful results. In a comparative study, the thermal (DSC/TGA) and luminescence properties of all the four compounds were investigated. The compounds of the $M^{II}(2,5\text{-DMT})$ composition show high thermal stability up to more than 300 °C, with the $Zn^{II}$ compound having the lowest decomposition temperature. $M^{II}(2,5\text{-DMT})$ with $M^{II} = Mn^{II}$, $Zn^{II}$ and **2** show a bright luminescence upon blue light irradiation ($\lambda = 405$ nm), whereas $Co^{II}$ in **1** quenches the emission. While $Zn^{II}$ in $Zn^{II}(2,5\text{-DMT})$ and $Mg^{II}$ in **2** do not significantly influence the (blue) emission and excitation bands compared to the free 2,5-DMT ligand, $Mn^{II}$ in $Mn^{II}(2,5\text{-DMT})$ shows an additional metal-centred red emission.

**Keywords:** coordination polymers; fluorescence; metal–organic frameworks; methoxy substituents; terephthalates

## 1. Introduction

Since their discovery more than 20 years ago [1,2], the research on MOFs (metal–organic frameworks) at the border between coordination, solid state, and materials chemistry has continued to attract an ever-increasing amount of interest. The simple design starting from a large variety of different metal cations or metal–oxo clusters as nodes and an almost unlimited number of possible organic linker ligands has now led to more than 100,000 entries in the MOF subset [3] of the CSD database [4], although not all entries follow strictly the recommendations of the IUPAC terminology for MOFs [5]. Since their first introduction, many possible applications in the fields of gas storage [6] and gas separation/purification [7], catalysis [8], drug transport/delivery [9], sensing [10], or electronic applications such as proton [11] and lithium-ion conductivity [12] have been discussed for this class of materials.

With respect to sensing, fluorescent (or, more generally, luminescent) MOFs [13] are being focused upon as a changing wavelength or a diminishing signal upon uptake of an adsorbate is simple to detect. Luminescence can be either metal- or linker-based, including ligand-to-metal charge transfer (LMCT) and metal-to-ligand charge transfer (MLCT) processes [14]. For the former, lanthanide-based MOFs are most prominent, e.g., $Eu^{3+}$-based materials with a strong red or $Tb^{3+}$-based compounds with a strong green luminescence [15]. There is an almost uncountable number of publications in this field, but it is our impression that linker-based luminescence has rarely been investigated, although even simple and very frequently used conjugated linkers such as 1,3,5-benzenetricarboxylic acid (BTC) or 1,4-benzenedicarboxylic acid (BDC) show a weak emission at 440 nm [16] and

388 nm [17], respectively. For potential sensing applications, linker-based luminescence holds many promising perspectives due to a direct interaction between the linker and the adsorbate, whereas a luminescent metal cation is somewhat "hidden" in its coordination sphere, making the influence of a non-coordinating adsorbate on its emission properties apparently weaker.

As a very spectacular result, linker-based luminescence was applied to detect defects within crystalline MOFs with a high spatial resolution [18]. In this work, based on MOFs with the UiO-67 topology, the authors used bulky fluorescein isothiocyanate or rhodamine B isothiocyanate substituents for their approach. However, there are also linkers with much smaller substituents, which show a strong luminescence. Among them, 2,5-dimethoxyterephthalic acid (2,5-DMT) shows a strong blue emission at $\lambda_{em}$ = 410 nm ($\lambda_{ex}$ = 370 nm)[this work], which compares well with the published results on its dimethyl ester ($\lambda_{em}$ = 402 nm with $\lambda_{ex}$ = 320 nm) [19]. In the literature, several coordination polymers (CP) and MOFs have already been reported with the 2,5-DMT linker and the $Mn^{2+}$ [20], $Zn^{2+}$ [20–22], $Co^{2+}$ [23], $Cu^{2+}$ [24], $Th^{4+}$ [25], and $Eu^{3+}$ metal cations [26]. To our surprise, the luminescence properties of the resulting materials were not investigated in most of these reports. The only exception is the $Eu^{3+}$-based compound (doped with $Tb^{3+}$), where a mainly lanthanide-based emission was observed [26]. Mertens et al. reported solvent-free coordination polymers $M^{II}$(2,5-DMT) with $M^{II}$ = $Mn^{2+}$, $Zn^{2+}$ [20]. In the following, we add $Co^{II}$(2,5-DMT) to this series and compare the thermal stability as well as the luminescence behaviour of all the three compounds. Additionally, we present the first $Mg^{2+}$-based MOF of the $Mg_2$(2,5-DMT)$_2$(DMF)$_2$ composition, which is discussed in the context of the three aforementioned CPs.

## 2. Materials and Methods

### 2.1. Synthesis of the Linker

All reagents were purchased commercially and used without further purification.

General: The synthesis of 2,5-dimethoxy terephthalic acid mainly followed the protocol of a two-step synthesis provided in the literature [27], but with an increased reaction time of the first step to increase the overall yield.

Synthesis of diethyl 2,5-dimethoxybenzene-1,4-dicarboxylate: The compound was synthesised in dry glassware under an inert (argon) atmosphere: 1.337 g (5.26 mmol, 1.00 eq.) diethyl-2,5-dihydroxy-terephthalate and 2.326 g (16.8 mmol, 3.20 equiv.) potassium carbonate were dissolved in 16 mL dry acetone; 1.1 mL (2.508 g, 17.7 mmol, 3.36 eq.) $CH_3I$ was added and the suspension was stirred for 48 h at 60 °C. The solvent was evaporated under reduced pressure and the resulting residue was dissolved in water and poured into a separation funnel. After the recombination of the aqueous solutions, they were re-extracted with ethyl acetate. The organic layers were collected and dried through the addition of $MgSO_4$. Filtration through a glass funnel and evaporation of the solvent under reduced pressure led to a colourless powder with a yield of 93% (1.38 g, 4.90 mmol).

$^{1}$H-NMR: 300 MHz, $CDCl_3$: ppm = 7.37 (s, 2H, H-3/6), 4.39 (q, J = 7.1 Hz, 4H, H-8/11), 3.89 (s, 6H, H-13/14), 1.40 (t, J = 7.1 Hz, 6H, H-9/12).

$^{13}$C-NMR: 75 MHz, $CDCl_3$: ppm = 165.7 (C-7/10), 152.5 (C-2/5), 124.5 (C-1/4), 115.5 (C-3/6), 61.5 (C-8/11), 57.0 (C-13/14), 14.4 (C-9/12).

### 2.2. Synthesis of Coordination Polymers/MOFs

$Co^{II}$(2,5-DMT) (**1**): 19.9 mg 2,5-dimethylterephthalic acid (0.087 mmol, 1.00 eq.) and 16.6 mg $Co(NO_3)_2$·6 $H_2O$ (0.090 mmol, 1.03 eq.) were dissolved in DMF (2.7 mL) in a glass vial (10 mL). The vial was sealed and heated at 100 °C for 48 h.

$Mn^{II}$(2,5-DMT): 20.3 mg 2,5-dimethylterephthalic acid (0.089 mmol, 1.00 eq.) and 15.8 mg $Mn(NO_3)_2$·4 $H_2O$ (0.88 mmol, 0.99 eq.) were dissolved in DMF (2.7 mL) in a glass vial (10 mL). The vial was sealed and heated at 100 °C for 48 h.

Zn$^{II}$(2,5-DMT): 19.4 mg 2,5-dimethylterephthalic acid (0.085 mmol, 1.00 eq.) and 16.7 mg Zn(NO$_3$)$_2$·6 H$_2$O (0.088 mmol, 1.03 eq.) were dissolved in DMF (2.7 mL) in a glass vial (10 mL). The vial was sealed and heated at 100 °C for 48 h.

Mg$_2$(2,5-DMT)$_2$(DMF)$_2$ (**2**): 17.1 mg 2,5-dimethylterephthalic acid (0.075 mmol, 1.00 eq.) and 12.2 mg Mg(NO$_3$)$_2$·6 H$_2$O (0.082 mmol, 1.09 eq.) were dissolved in DMF (2.7 mL) in a glass vial (10 mL). The vial was sealed and heated at 100 °C for 48 h.

### 2.3. Analytical Methods

PXRD patterns were recorded on a Rigaku MiniFlex 600-C diffractometer (Cu K$\alpha$ radiation, Ni filter) (Tokyo, Japan) as flat samples. The typical recording times were 30 min. The obtained data were analysed and processed with the WinXPow programme package [28]. Additionally simulated patterns were generated from the Crystallographic Information Files of the respective substances with WinXPow [28]. The resulting data from the measurements were visualised with Gnuplot [29].

DSC/TG measurements were conducted on a Mettler Toledo (Gießen, Germany) DSC1 coupled with a TGA/DSC1 (Star System). The measured samples were placed and weighed in a corundum crucible under an argon stream (40 mL/min). Initially, the sample was heated to 30 °C and held at this temperature for 10 min. Subsequently, the sample was heated to 1000 °C with a heating rate of 10 °C/min. Finally, the data were evaluated with the STARe program package.

Luminescence measurements were carried out using a FLS980 spectrometer from Edinburgh Instruments (Livingston, UK) with a xenon lamp and a PMT detector. The measurements in the solution (2,5-DMT) were carried out in DMF using quartz glass cuvettes, the solid state measurements of 2,5-DMT and **2**—between two quartz glass plates. Mn$^{II}$(2,5-DMT) and Zn$^{II}$(2,5-DMT) were measured as KBr pellets. All the measurements were conducted at room temperature.

### 2.4. X-ray Single-Crystal Structure Analysis

The measurements were carried out on a Bruker D8 Venture diffractometer with either Ag K$\alpha$ ($\lambda$ = 0.56086 Å) or Mo K$\alpha$ ($\lambda$ = 0.71073 Å) radiation and a multi-layered mirror monochromator. For the reduction of the diffraction data by integration and absorption correction, the SAINT [30] and SADABS/TWINABS [31,32] programs from the APEX4 program package [33] were used. The determination of the space group and the starting model was carried out with the SHELXT program [34]. For further refinement, the SHELXL-18 [35] program was applied using the least squares method. All non-hydrogen atoms were refined with anisotropic displacement parameters. The hydrogen atoms were refined isotropically on the calculated positions using a riding model with their U$_{iso}$ values constrained to 1.5 times the U$_{eq}$ of their pivot atoms for terminal sp$^3$ carbon atoms and 1.2 times for all other carbon atoms. The data obtained from the SCXRD measurement of **2** revealed a non-merohedral crystal twin whose two domains were tilted by 4° with respect to each other. These two domains were integrated separately and subjected to twin absorption correction. Based on the reflections of the more dominant domain, an HKLF 4 file was generated, from which the structure was initially solved and refined. The final structure model was refined against the reflections of both domains (HKLF 5).

### 2.5. Further Software Programs

Visualisations of all crystal structures were made with the Diamond 4.6 program package [36]. Visualisation of the fluorescent data and DSC/TGA measurements was completed using the Origin 8.5 program package [37]. The ChemDraw Professional 15.0 program package [38] was used to visualise the organic molecules.

## 3. Results and Discussion

### 3.1. Synthesis and General Characterisation

Through a solvothermal reaction of (1:1) 2,5-dimethoxyterephthalic acid (2,5-DMT) and the respective metal nitrates ($M^{II}(NO_3)_2 \cdot x$ $H_2O$ with $M^{II}$ = Co, Mg, Zn, x = 6 and $M^{II}$ = Mn, x = 4), coordination polymers $M^{II}$(2,5-DMT) with $M^{II}$ = Co (**1**), Mn, Zn and the $Mg_2$(2,5-DMT)$_2$(DMF)$_2$ MOF (**2**) were obtained. The reactants were thoroughly mixed, dissolved in dimethylformamide (DMF), and heated for 48 h at 100 °C in a 10 mL glass vial. The resulting precipitates were filtered and dried. Powder X-ray diffractograms (PXRD, Figures S1–S4, Supporting Information) revealed that the $M^{II}$(2,5-DMT) coordination polymers with $M^{II}$ = $Mn^{II}$, $Zn^{II}$ (*C*2/*c*, Z = 4) [20] known from the literature and the new $Co^{II}$ compound **1** crystallise isostructurally to each other. The PXRD patterns confirm that all the three compounds were obtained as samples with a high degree of purity (Table 1). The PXRD pattern of **2**, however, looks completely different to the other three, thus revealing that a material with a different structural arrangement was synthesised. After solving its crystal structure (*P*1, Z = 2; vide infra), it became evident that a MOF-type structure with DMF-filled channels was formed. Again, a very good correlation between the experimental PXRD pattern and the pattern simulated from the solved crystal structure confirms a high purity of this material. Single crystals of **1** and **2**, which exhibit a block-shaped habitus, were isolated from the precipitates mentioned above and measured on an X-ray single-crystal diffractometer. The crystals of **1** are transparent pinkish violet, whereas the crystals of **2** are—as expected—colourless.

**Table 1.** Calculated/observed values of the elemental analysis of $M^{II}$(2,5-DMT) with $M^{II}$ = $Co^{II}$ (**1**), $Mn^{II}$, $Zn^{II}$ and $Mg_2$(2,5-DMT)$_2$(DMF)$_2$ (**2**) with an error tolerance of ±0.3%.

|   | $Co^{II}$(2,5-DMT) (1) | $Mn^{II}$(2,5-DMT) | $Zn^{II}$(2,5-DMT) | $Mg_2$(2,5-DMT)$_2$(DMF)$_2$ (2) |
|---|---|---|---|---|
| N | −/0.21 | −/0.08 | −/0.20 | 4.36/4.40 |
| C | 42.43/42.76 | 43.03/43.20 | 41.48/41.55 | 48.56/48.20 |
| H | 2.85/2.84 | 2.89/2.81 | 2.79/2.70 | 4.70/4.73 |
| S | −/− | −/− | −/− | −/− |

### 3.2. Crystal Structures

Complete crystallographic data can be found in the Supporting Information (Tables S1–S10). The three isostructural coordination polymers crystallise in the monoclinic space group *C*2/*c* [20]; the novel Mg MOF crystallises in the triclinic space group *P*1. For comparison, selected crystallographic data of all the four compounds are given in Table 2.

The $Co^{II}$ CP (**1**) crystallises isostructurally to the known $Mn^{II}$ and $Zn^{II}$ CPs, which have already been described in the literature [20]. Therefore, the crystal structure of **1** is only briefly discussed. Its asymmetric unit (ORTEP plot) is given as Figure S5 in the Supporting Information. In these compounds, the $M^{II}$ cation is coordinated by four oxygen atoms stemming from the carboxylate groups of four different 2,5-DMT linkers (four shorter $M^{II}$–O distances given in Table 2). This leads to a distorted tetrahedral coordination, which is depicted for **1** in Figure 1. To quantify the distortion of the coordination spheres, we used the continuous shape measures approach (CShM) [40]; the respective values for a tetrahedral coordination (T-4) are given in Table 2. Values much larger than 1 indicate a severe distortion of the tetrahedral coordination. In Figure 1, two further oxygen atoms with significantly longer $M^{II}$–O bonds (cp. Table 2) are depicted stemming from the methoxy groups of two 2,5-DMT linkers. Thus, a distorted octahedral coordination can be assumed, but CShM values (OC-6) >> 1 again indicate a strong distortion. However, it is remarkable (cp. Table 2) that a decreasing distortion of the tetrahedral coordination from the Co compound to the Mn and the Zn compounds goes along with an increasing distortion of the octahedral coordination. Obviously, with these methoxy substituents in terephthalate-based linkers, pockets are formed to accommodate metal cations. The size of

the metal cations seems to have a direct influence on the pocket's shape, more tetrahedral or more octahedral. This is also reflected in the $Co^{II}$–O distances of **1**, which show a large spread from 1.9834(7) Å to 2.3533(7) Å, while in the $Co^{II}$ CPs with terephthalate ligands, a less distorted octahedral coordination with $Co^{II}$–O distances from 2.059(3) Å to 2.119(9) Å is found [41,42].

**Table 2.** Selected crystallographic data of $M^{II}$(2,5-DMT) with $M^{II}$ = $Co^{II}$ (**1**), $Mn^{II}$, $Zn^{II}$ and $Mg_2$(2,5-DMT)$_2$(DMF)$_2$ (**2**).

| | $Co^{II}$(2,5-DMT) (**1**) | $Mn^{II}$(2,5-DMT) | $Zn^{II}$(2,5-DMT) | $Mg_2$(2,5-DMT)$_2$(DMF)$_2$ (**2**) |
|---|---|---|---|---|
| Crystal system | monoclinic | monoclinic | monoclinic | triclinic |
| Space group, Z | $C2/c$, 4 | $C2/c$, 4 | $C2/c$, 4 | $P1$, 2 |
| a/Å | 16.1305(5) | 16.7686(6) | 16.5936(6) | 8.833(3) |
| b/Å | 8.6024(3) | 8.4646(3) | 8.4438(3) | 9.691(3) |
| c/Å | 7.3426(2) | 7.4464(3) | 7.4838(3) | 18.674(6) |
| α/° | 90 | 90 | 90 | 98.274(7) |
| β/° | 96.425(1) | 99.093(1) | 97.649(2) | 93.305(10) |
| γ/° | 90 | 90 | 90 | 107.308(9) |
| Volume/Å$^3$ | 1012.47(5) | 1043.66(7) | 1039.25(7) | 1501.6(8) |
| Temp./K | 100(2) | 153(2) | 153(2) | 100(2) |
| Ionic radius, CN = 4 [39] | 0.72 Å ($Co^{2+}$, hs) | 0.80 Å ($Mn^{2+}$, hs) | 0.74 Å ($Zn^{2+}$) | 0.71 Å ($Mg^{2+}$) |
| $M^{II}$–O/Å | 1.9834(7), 2× 2.0389(7), 2× 2.3532(7), 2× | 2.0761(5), 2× 2.1391(6), 2× 2.5595(6), 2× | 1.9547(13), 2× 2.0023(13), 2× 2.6223(14), 2× | Mg1: 1.961(3), 1.989(3), 2.007(3), 2.118(3), 2,198(3), 2.284(3) Mg2: 2.042(4), 2.048(4), 2.080(3), 2.082(3), 2.100(3), 2.107(3) |
| CShM values [40] | 4.896 (T-4) 1.697(OC-6) | 3.847 (T-4) 2.848 (OC-6) | 2.428 (T-4) 3.280 (OC-6) | Mg1: 2.915 (OC-6) Mg2: 0.145 (OC-6) |
| Ref. | CCDC-2225418[this work] | CCDC-813469 [20] | CCDC-813470 [20] | CCDC-2225419[this work] |

It should be noted that Mertens et al. chose a different description of their $Mn^{II}$ and $Zn^{II}$ CPs as they included two even more distant oxygen atoms, which led to CN = 8 and a distorted "super dodecahedron" [20]. None of these descriptions can be considered wrong or correct, they are more an expression of the flexibility of metal coordination in such compounds with 2,5-DMT ligands. $M^{II}O_n$ polyhedra are connected to chains running along [001], which are interconnected through 2,5-DMT linkers to form a 3D coordination network. The resulting topology shows some similarities with MOFs of the MIL series (sra topology). For a more detailed description of these crystal structures, see [20]. Neither Mertens et al. nor our group found an indication of permanent porosity in these materials (cp. Figure S6, Supporting Information, showing a space-filling representation of **1**).

Using the same reaction conditions that led to the formation of $M^{II}$(2,5-DMT) with $M^{II}$ = $Co^{II}$ (**1**), $Mn^{II}$, $Zn^{II}$, we obtained a completely different compound when $Mg(NO_3)_2 \cdot 6\ H_2O$ was used as the starting material. It was shown that this structure was also formed when the cooling time was increased to 96 h and/or when lauric acid was added as a monocarboxylic additive to improve the crystallinity. The resulting crystal structure is shown in Figure 2. The asymmetric unit of $Mg_2$(2,5-DMT)$_2$(DMF)$_2$ (**2**) consists of two crystallographically distinguishable magnesium atoms (Mg1, Mg2), one complete and two half-linker anions as well as two coordinating DMF molecules.

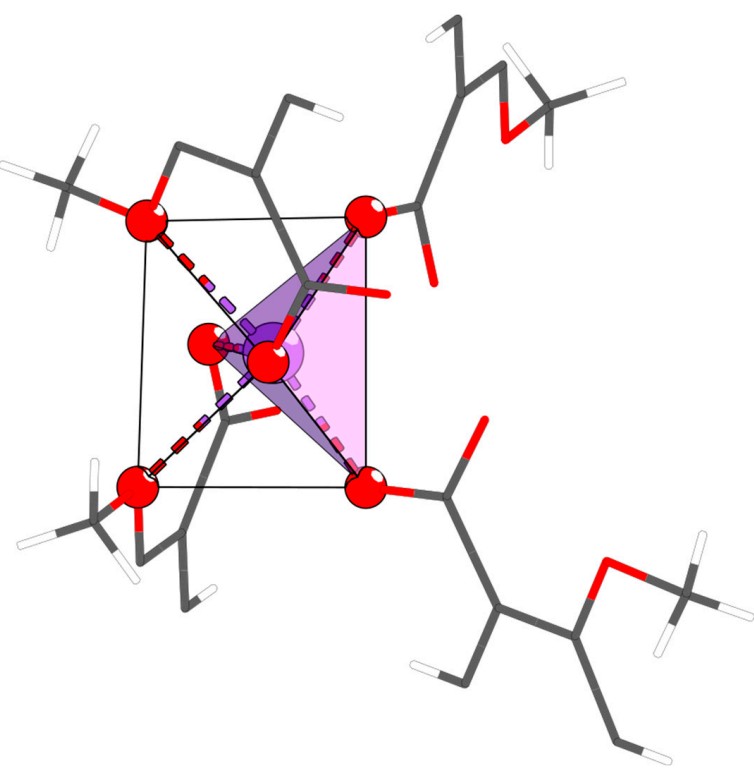

**Figure 1.** The coordination sphere of Co$^{II}$ in the crystal structure of Co$^{II}$(2,5-DMT), **1**; Co (purple sphere), O (red spheres), C (grey wireframes), and H (white wireframes). A tetrahedral Co$^{II}$O$_4$ coordination is emphasized by purple shading, an extension to an octahedral Co$^{II}$O$_6$ coordination—by thin dark grey lines.

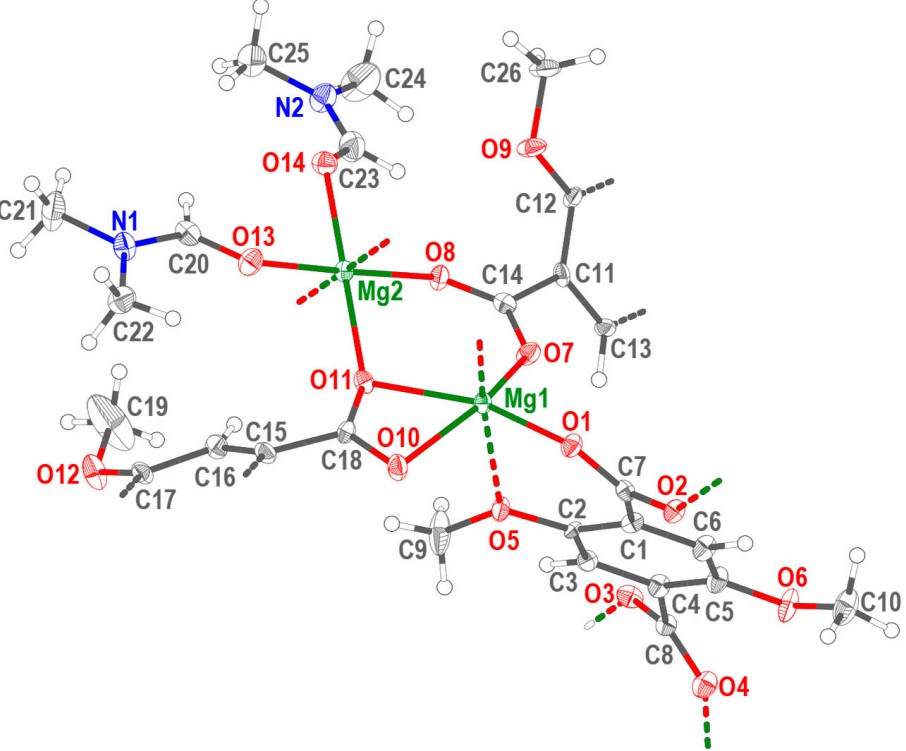

**Figure 2.** The asymmetric unit of Mg$_2$(2,5-DMT)$_2$(DMF)$_2$ (**2**) with atomic numbering; thermal ellipsoids are drawn with a 50% probability.

Both Mg atoms form distorted octahedral MgO$_6$ coordination spheres, which, however, show large differences. Mg1 is coordinated by five oxygen atoms which stem from the carboxylate groups of four different 2,5-DMT ligands (one carboxylate group coordinates in a bidentate chelating mode). The sixth oxygen atom belongs to a methoxy group of one of the four 2,5-DMT linkers. The resulting octahedron shows a severe distortion, as expressed by the large CShM value of 2.915 (Table 2). This confirms, as found for M$^{II}$(2,5-DMT) with M$^{II}$ = Co$^{II}$ (**1**), Mn$^{II}$, Zn$^{II}$, that, again, pockets are formed including the methoxy substituent, which, due to spatial restrictions, leads to distorted polyhedra if the metal cations do not fit perfectly into these pockets. In contrast, Mg2 is coordinated by four oxygen atoms of the carboxylate groups of four different 2,5-DMT anions and two oxygen atoms of two different DMF molecules. As there are no spatial restrictions, i.e., the oxygen atoms can be freely arranged, an almost undistorted octahedron is observed with a small CShM value (0.145). This is also reflected in the Mg–O distances: the Mg1–O distances range from 1.961(3) Å (O1) to 2.284(3) Å (O5), the Mg2–O distances—from 2.042(4) Å (O2) to 2.107(3) Å (O14). As expected, the longest bond is found between Mg1 and the oxygen atom O5 of the methoxy group. It is more than 0.1 Å longer than the typical Mg–O bond lengths found in the literature [43,44].

O11 bridges both MgO$_6$ octahedra, thus forming a corner-connected dimer. These dimeric units are interconnected through the 2,5-DMT ligands creating a three-dimensional network (Figure 3). This network forms channels into which the coordinating DMF molecules protrude. The pore sizes were calculated with the PLATON program package [45] taking the respective van der Waals radii into account, resulting in diameters from 3.10 Å to 4.64 Å. The channels (Figure 4) penetrate the whole framework of **2** in a wave-like fashion. It was assumed that these voids might be accessible to guest uptake after removal of the coordinating DMF molecules. This will be discussed in more detail below.

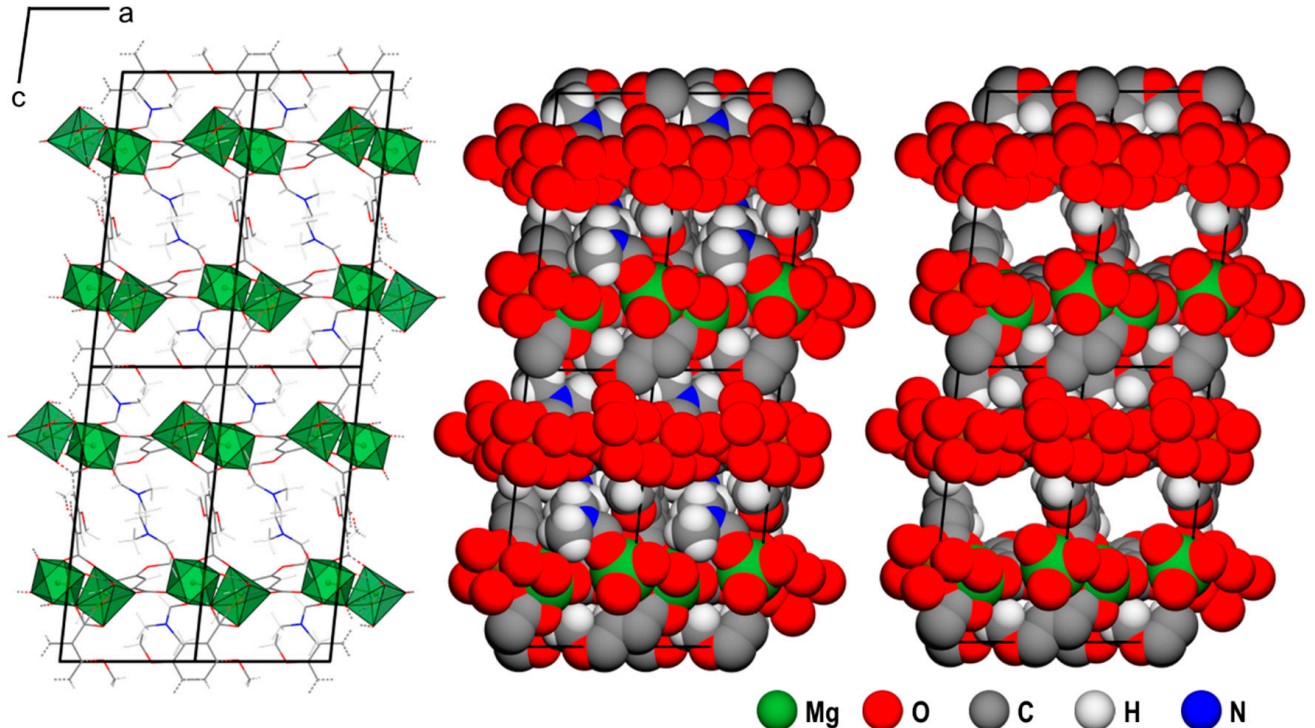

**Figure 3.** Section of the crystal structure of Mg$_2$(2,5-DMT)$_2$(DMF)$_2$ (**2**) in a view along [010]. (**Left**): representation of the linker anions as wireframes, the MgO$_6$ octahedra are emphasized in green; (**middle**): space-filling representation considering the van der Waals radii with the DMF molecules; (**right**): space-filling representation without the DMF molecules.

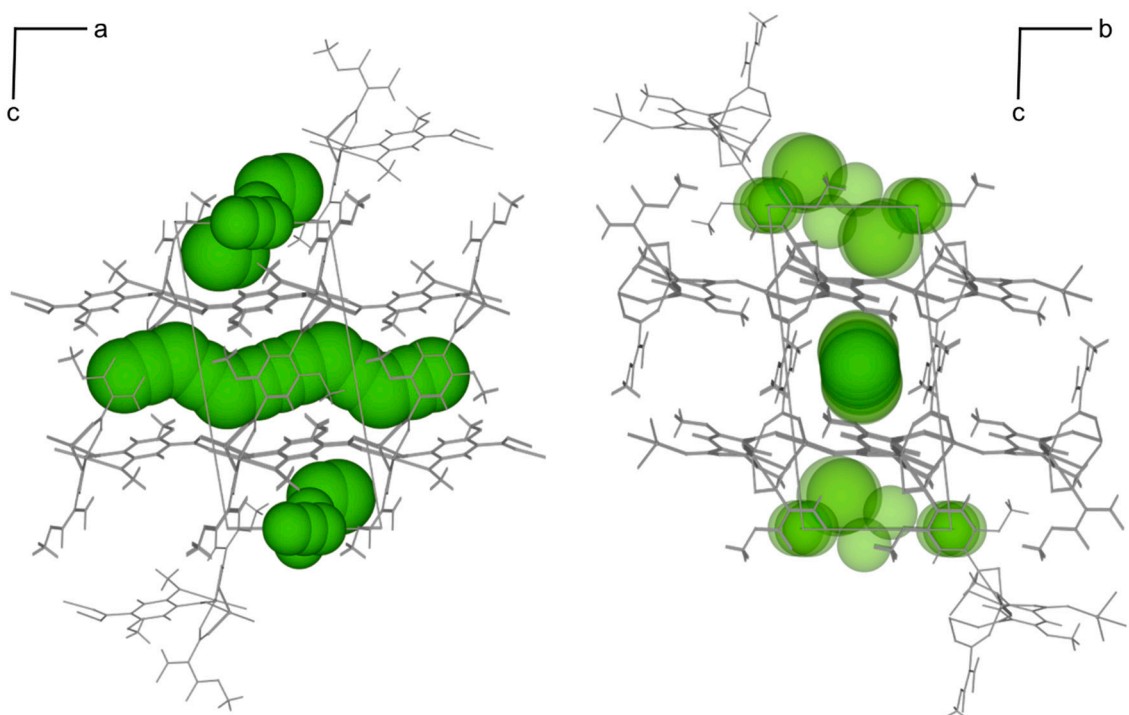

**Figure 4.** Visualisation of possible voids in Mg$_2$(2,5-DMT)$_2$(DMF)$_2$ (**2**) if the coordinating solvent molecules (DMF) can be removed; projections along [010] (**left**) and [100] (**right**).

It is remarkable that for Mg$^{2+}$, a completely different structure is observed when compared to Co$^{2+}$, Mn$^{2+}$, and Zn$^{2+}$. This might be explained by the smallest ionic radius of Mg$^{2+}$ compared to the other three 3d metal cations Minor variations of the reaction conditions always lead to materials with the crystal structure of **2**. The major difference between the two different structure types is that there is no coordination of the solvent molecules in any of the three Co$^{2+}$, Mn$^{2+}$, and Zn$^{2+}$ coordination polymers, whereas two DMF molecules coordinate to Mg$_2$ in the magnesium compound. This leads to small voids and wave-like channels with a potentially accessible porosity (vide infra).

### 3.3. Thermogravimetric Analyses

All the compounds M$^{II}$(2,5-DMT) with M$^{II}$ = Co$^{II}$ (**1**), Mn$^{II}$, Zn$^{II}$ and **2** were investigated with regard to their thermal stability by means of coupled thermogravimetry (TG) and differential scanning calorimetry (DSC) using an inert Ar atmosphere and a 10 °C min$^{-1}$ heating rate.

The TG curve of Mg$_2$(2,5-DMT)$_2$(DMF)$_2$ (**2**) shows four separated weight losses up to ~1000 °C (Figure 5, DSC curves are given as Figure S7, Supporting Information). The first mass loss of 4.64% was detected between 50 °C and 140 °C. It was followed by the second one between 140 °C and 250 °C. Calculations show that the sum of both fits almost perfectly to the release of the two coordinating DMF molecules (calc.: 22.7%, detected: 23.4%). The two following mass losses describe the decomposition of the framework starting above 300 °C.

For the three isostructural coordination polymers M$^{II}$(2,5-DMT) with M$^{II}$ = Co$^{II}$ (**1**), Mn$^{II}$, Zn$^{II}$, a very similar thermal stability was found (Figure 5). For Zn$^{II}$(2,5-DMT), the first mass loss with 30.1% occurred between 280 °C and 430 °C, which would fit a total decarboxylation of the linker molecule (calc.: 30.4%). Two further decreases were observed at higher temperatures between 430–550 °C and 550–800 °C, respectively. The first decrease could correspond to the release of a methanol molecule (obs.: 11.09%, calc.: 11.06%), which is formed from the methoxy group. Unfortunately, the amount of the residue after heating to 1000 °C was too small to record a PXRD pattern. In the case of Mn$^{II}$(2,5-DMT), a

TGA curve very similar to that of the Zn compound was observed (Figure 5). The first mass decreases were detected between 340–440 °C and 440–575 °C, i.e., at slightly higher temperatures compared to the Zn CP. The agreement between the calculated and observed values for a full decarboxylation and the cleavage of a methanol molecule is not as good as for the Zn compound (obs.: 43.02%, calc.: 38.4%). Co$^{II}$(2,5-DMT) (**1**) also shows thermal stability up to at least 300 °C. Here, the first mass loss occurred at ~430 °C, very similar to the Mn$^{II}$ compound (obs.: 28.82%; calc.: 31.09% for a complete decarboxylation). Further decreases occurred between 420–510 °C and 510–630 °C, most likely due to the cleavage of the methoxy groups.

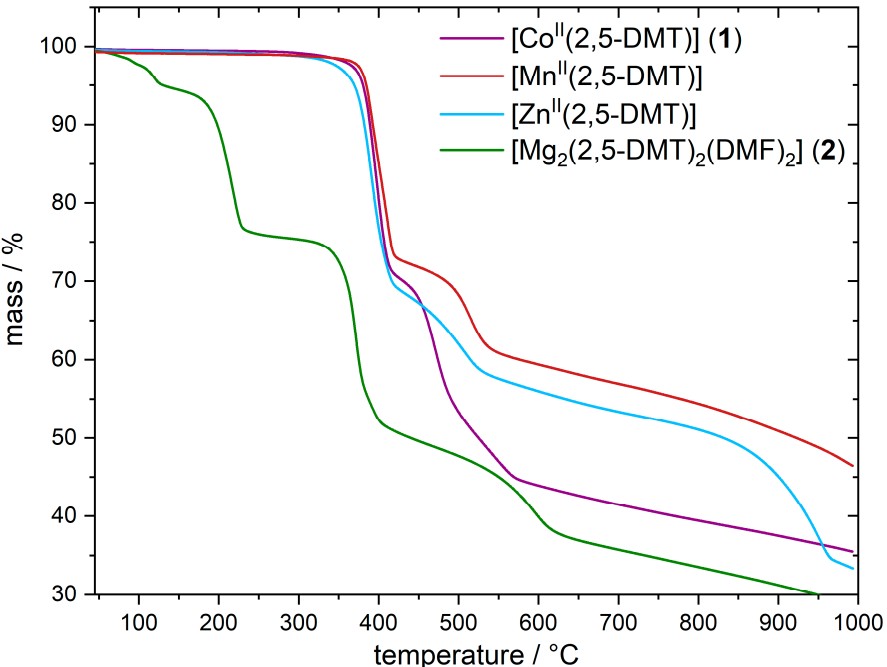

**Figure 5.** Comparison of the measured TG curves of Co$^{II}$(2,5-DMT) (violet curve, **1**), Mn$^{II}$(2,5-DMT) (red curve), Zn$^{II}$(2,5-DMT) (blue curve), and Mg$_2$(2,5-DMT)$_2$(DMF)$_2$ (green curve, **2**).

Comparison of all the M$^{II}$(2,5-DMT) compounds with M$^{II}$ = Co$^{II}$ (**1**), Mn$^{II}$, Zn$^{II}$ shows that Mn$^{II}$(2,5-DMT) has the highest thermal stability, followed by Co$^{II}$(2,5-DMT) (**1**) with an only slightly decreased decomposition temperature, whereas Zn$^{II}$(2,5-DMT) shows a significantly lower thermal stability, by approx. 20 °C. As expected, the lowest thermal stability was found for Mg$_2$(2,5-DMT)$_2$(DMF)$_2$ (**2**), which is due to the release of the coordinating solvent molecules (DMF) upon heating. Remarkably, above 300 °C, the TG curve of **2** shows a very similar trend to the one found for the three solvent-free CPs starting at room temperature. It is therefore suggested that a coordination polymer Mg(2,5-DMT) isostructural to M$^{II}$(2,5-DMT) with M$^{II}$ = Co$^{II}$ (**1**), Mn$^{II}$, Zn$^{II}$ (*C2/c*, *Z* = 4) might be formed after the release of the two DMF molecules. To confirm this assumption, a sample of **2** was heated at 280 °C in an argon stream for one hour. However, the PXRD pattern of the resulting material shows the same reflections as observed at room temperature, i.e., no structural change occurred under these conditions. Only the crystallinity of the material decreased significantly. Obviously, the framework of **2** collapses upon the release of the coordinating DMF guests. This also explains why we were unable to activate **2** and record a type I isotherm in the N$_2$ gas sorption measurements.

### 3.4. Luminescence Properties

Since coordination polymers M$^{II}$(2,5-DMT) with M$^{II}$ = Co$^{II}$ (**1**), Mn$^{II}$, Zn$^{II}$ are isostructural compounds, the comparison of their emission and excitation bands could allow assumptions about the influence of different metal cations on the respective luminescence

properties. The measured emission and excitation spectra are given in Figure 6 and compared with the respective spectra of the free 2,5-DMT ligand (grey reference). Table 3 summarises the optical properties of $M^{II}$(2,5-DMT) with $M^{II}$ = $Mn^{II}$, $Zn^{II}$, **2**, and the free ligand. For $Co^{II}$(2,5-DMT) (**1**), no emission was observed as it was quenched by the $Co^{2+}$ cations as known from the literature [46]. Pictures of the excited $M^{II}$(2,5-DMT) compounds with $M^{II}$ = $Co^{II}$ (**1**), $Mn^{II}$, $Zn^{II}$ and **2** after blue light irradiation are shown in the Supporting Information (Figure S9).

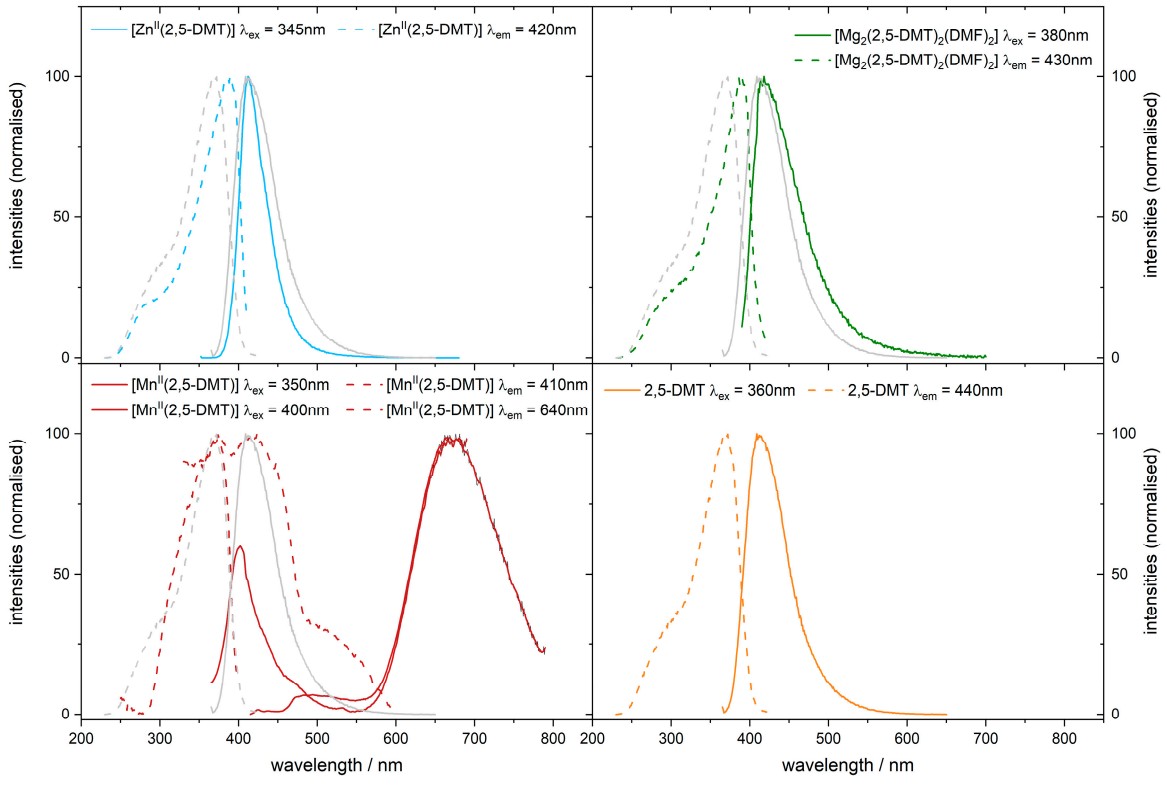

**Figure 6.** The emission (solid trace) and excitation spectra (dotted trace) of $Zn^{II}$(2,5-DMT) (**top left**, blue traces), $Mg_2$(2,5-DMT)$_2$(DMF)$_2$, **2** (**top right**, green traces), $Mn^{II}$(2,5-DMT) (**bottom left**, red traces), and 2,5-dimethoxyterephthalic acid (**bottom right**, orange traces) measured at 295 K. The emission and excitation spectra of the free 2,5-DMT linker are additionally plotted as a reference (grey traces) in each spectrum of the coordination polymers.

**Table 3.** Maximum excitation, emission and absorption wavelengths of $M^{II}$(2,5-DMT) with $M^{II}$ = $Mn^{II}$, $Zn^{II}$, **2**, and the free 2,5-DMT ligand (s = shoulder).

| | Max. Excitation/nm | Max. Emission/nm | Max. Absorption/nm |
|---|---|---|---|
| 2,5-DMT | 370 | 410 | 220, 250(s), 360 |
| $Mn^{II}$(2,5-DMT) | 370, 420 | 400, 660 | 200, 260(s), 370(s) |
| $Zn^{II}$(2,5-DMT) | 390 | 410 | 205, 260(s), 390(s) |
| $Mg_2$(2,5-DMT)$_2$(DMF)$_2$, **2** | 390 | 420 | 205, 250(s), 325 |

For $Zn^{II}$(2,5-DMT) and $Mg_2$(2,5-DMT)$_2$(DMF)$_2$ (**2**), the emission and excitation bands are similar to those observed for the free 2,5-DMT ligand. The excitation bands of both compounds are slightly red-shifted compared to the ligand's excitation band due to a weak interaction with the metal centre, while the emission band centres are almost identical for the coordination compounds and the ligand. This is consistent with the measured absorption spectra (Table 3), where the absorption bands of $Zn^{II}$(2,5-DMT) and $Mg_2$(2,5-DMT)$_2$(DMF)$_2$ (**2**) do not significantly differ from the absorption of the free ligand. Therefore, we attribute

the emission bands in **2** and $Zn^{II}$(2,5-DMT) to a ligand-centred transition, while the metals do not participate in the radiative pathway in a significant way.

The ligand-centred emission band is also identified for the $Mn^{II}$(2,5-DMT) coordination polymer. In addition, a second emission band at 660 nm is observed after excitation at 410 nm, resulting in a red emission of the coordination polymer (cp. Figure S9). This emission band is attributed to a metal-centred transition of the $Mn^{2+}$ ion as it has not been observed for the $Zn^{2+}$ and $Mg^{2+}$ coordination compounds and the free ligand. Through the red emission, it is possible to make an additional statement about the coordination, which cannot be unambiguously identified via X-ray single-crystal structure analysis as discussed above. While tetrahedrally coordinated $Mn^{2+}$ emits green light, a red-light emission is observed for octahedrally coordinated $Mn^{2+}$ ions caused by a transition from the excited $^4T_{1g}(^4G)$ state to the $^6A_{1g}(^6S)$ ground state [47,48]. Therefore, from these UV–vis spectra, one can conclude that the coordination sphere of $Mn^{2+}$ in $Mn^{II}$(2,5-DMT) is best described as an $MnO_6$ octahedron.

## 4. Conclusions

We synthesized a new coordination polymer $Co^{II}$(2,5-DMT) (**1**) containing a fluorescent 2,5-DMT (2,5-dimethoxyterephthalate) linker; **1** is isostructural to the known $Mn^{II}$ and $Zn^{II}$ congeners. Attempts to synthesize an $Mg^{2+}$ analogue led to the synthesis of $Mg_2$(2,5-DMT)$_2$(DMF)$_2$ (**2**) with coordinating DMF molecules and a MOF-type structure. Attempts to remove the DMF guests upon heating failed so that no permanent porosity could be proven. In a comparative study, thermal stability of all the four compounds was investigated. The solvent-free $M^{II}$(2,5-DMT) coordination polymers with $M^{II}$ = $Co^{II}$ (**1**), $Mn^{II}$, and $Zn^{II}$ showed a very similar decomposition behaviour, with the $Zn^{II}$ compound being a slightly less stable material. Nonetheless, all the three compounds decomposed clearly above 300 °C. $M^{II}$(2,5-DMT) with $M^{II}$ = $Mn^{II}$, $Zn^{II}$, and **2** as well as the pristine linker 2,5-DMT exhibited a strong emission upon irradiation with blue/UV light, while in the $Co^{2+}$-containing material (**1**), the emission was quenched. $Zn^{II}$(2,5-DMT) and $Mg_2$(2,5-DMT)$_2$(DMF)$_2$ (**2**) showed a mainly ligand-based blue emission, while for $Mn^{II}$(2,5-DMT), an additional metal-based red emission was found. The latter points to an octahedral coordination of $Mn^{2+}$. This is remarkable from a structural point of view, as in solvent-free $M^{II}$(2,5-DMT) compounds, pockets around the $M^{2+}$ cations are formed with an inner (distorted tetrahedral) and an outer coordination sphere. The inner sphere is solely formed by four oxygen atoms of carboxylate groups, whereas in the outer sphere, two oxygen atoms of the methoxy groups are also included, leading to a distorted octahedral coordination. Although the $Mn–O_{methoxy}$ distances are distinctively longer than the $Mn–O_{carboxylate}$ distances (by more than 0.4 Å), the UV/vis spectra of $Mn^{II}$(2,5-DMT) clearly indicate that there is still a significant $Mn–O_{methoxy}$ interaction.

We believe that the 2,5-DMT ligand is an attractive linker for the construction of luminescent coordination polymers and MOFs. Especially in MOFs with a permanent porosity, it might lead to interesting materials for sensing applications. The synthesis of such porous and luminescent MOFs is in the focus of our current research in this field.

**Supplementary Materials:** The following supporting information can be downloaded at: https://www.mdpi.com/article/10.3390/chemistry5020065/s1, Figure S1: PXRD pattern of [$Co^{II}$(2,5-DMT)] (**1**); Figure S2: PXRD pattern of [$Mn^{II}$(2,5-DMT)]; Figure S3: PXRD pattern of [$Zn^{II}$(2,5-DMT)]; Figure S4: PXRD pattern of [$Mg_2$(2,5-DMT)$_2$(DMF)$_2$] (**2**); Figure S5: Asymmetric unit of [$Co^{II}$(2,5-DMT)] (**1**); Figure S6: Space filling representation of [$Co^{II}$(2,5-DMT)] (**1**); Figure S7: DSC curves of [$Mn^{II}$(2,5-DMT)], [$Zn^{II}$(2,5-DMT)], [$Co^{II}$(2,5-DMT)] (**1**), and [$Mg_2$(2,5-DMT)$_2$(DMF)$_2$] (**2**); Figure S8: UV/vis measurements of 2,5-DMT and [$Mn^{II}$(2,5-DMT)], [$Zn^{II}$(2,5-DMT)], and [$Mg_2$(2,5-DMT)$_2$(DMF)$_2$] (**2**); Figure S9: Photographs of fluorescence of [$Mn^{II}$(2,5-DMT)], [$Zn^{II}$(2,5-DMT)], [$Co^{II}$(2,5-DMT)] (**1**), and [$Mg_2$(2,5-DMT)$_2$(DMF)$_2$] (**2**) upon blue light irradiation; Tables S1–S5: Crystallographic data of [$Co^{II}$(2,5-DMT)] (**1**); Tables S6–S10: Crystallographic data of [$Mg_2$(2,5-DMT)$_2$(DMF)$_2$] (**2**).

**Author Contributions:** A.E.L.C. synthesized all the compounds, conducted most of the measurements as well as analyses, and wrote parts of the original draft. L.S. helped A.E.L.C. with the luminescence measurements and wrote parts of that section. D.v.G. assisted A.E.L.C. in the single-crystal structure analysis. M.S.W. supervised L.S. and D.v.G. during their work. U.R. supervised the whole project, was in charge of administration and funding for the project, and wrote major parts of the original draft. All authors have read and agreed to the published version of the manuscript.

**Funding:** This research was funded by the Deutsche Forschungsgemeinschaft (DFG) under the grant number RU 546/12-1.

**Data Availability Statement:** Spectroscopic and DSC/TGA data as well as some crystallographic details are given in the electronic supplement. The crystal structures of compounds **1** and **2** were determined by single-crystal X-ray diffraction and the crystallographic data have been deposited with the Cambridge Crystallographic Data Centre as supplementary publication nos. CCDC-2225418 (**1**) and 2225419 (**2**). Copies of the data can be obtained free of charge on application to CCDC, 12 Union Road, Cambridge, CB21EZ (fax: (+44) 1223/336033; e-mail: deposit@ccdc.cam-ak.uk).

**Acknowledgments:** We acknowledge the help of C. Tobeck (DSC/TGA data acquisition) and R. Christoffels (gas sorption measurements).

**Conflicts of Interest:** The authors declare no conflict of interest.

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
