# Peer review of "Synthesis, Structure, and Spectroscopic Properties of Luminescent Coordination Polymers Based on the 2,5-Dimethoxyterephthalate Linker"

_chemistry, doi:10.3390/chemistry5020065_

Round 1

Reviewer 1 Report

In this manuscript, the authors report two new MOFs and compare the related properties with already reported Mn and Zn congeners. The topic itself is interesting and this paper might suitable for the publication on Chemistry, but before publication, revision should be made. More detail comments are as follows:

1. All the reported four MOFs are synthesized with closed experimental methods and processes. But, the last one, that is, Mg-based MOF show different structure. It is better to analyze the origin of this difference.

2. The final R indexes for Mg-based MOF is higher than that of Co-based one? Why? Is this due to the poor data or crystal quality?

In this manuscript, the authors report two new MOFs and compare the related properties with already reported Mn and Zn congeners. The topic itself is interesting and this paper might suitable for the publication on Chemistry, but before publication, revision should be made. More detail comments are as follows:

1. All the reported four MOFs are synthesized with closed experimental methods and processes. But, the last one, that is, Mg-based MOF show different structure. It is better to analyze the origin of this difference.

2. The final R indexes for Mg-based MOF is higher than that of Co-based one? Why? Is this due to the poor data or crystal quality?

Reviewer 2 Report

This paper reports on the structural characterization and luminescence properties of two new coordination polymers based on DMT linker.  This study shows clear results and deserves publication in Chemistry.  I have one comment the authors should consider. Since the originality of the Mn and Zn complexes belongs to the authors of reference 20, they should not be treated with similar numbering in this paper. Even in the table, it should be clearly distinguished from the Co and Mg complexes and treated as comparison targets.
